# Children’s Health in the Digital Age

**DOI:** 10.3390/ijerph17093240

**Published:** 2020-05-06

**Authors:** Birgitta Dresp-Langley

**Affiliations:** Centre National de la Recherche Scientifique, UMR 7357 ICube CNRS and Université de Strasbourg Hôpitaux Universitaires Faculté de Médecine, Pavillon Clovis Vincent, 4, rue Kirschleger, 67085 Strasbourg CEDEX, France; birgitta.dresp@unistra.fr

**Keywords:** digital environments, overexposure, children, daylight, vitamin D, melatonin, myopia, sleep loss, depression, obesity, internet addiction, serotonin, dopamine, oxidative stress

## Abstract

Environmental studies, metabolic research, and state of the art research in neurobiology point towards the reduced amount of natural day and sunlight exposure of the developing child, as a consequence of increasingly long hours spent indoors online, as the single unifying source of a whole set of health risks identified worldwide, as is made clear in this review of currently available literature. Over exposure to digital environments, from abuse to addiction, now concerns even the youngest (ages 0 to 2) and triggers, as argued on the basis of clear examples herein, a chain of interdependent negative and potentially long-term metabolic changes. This leads to a deregulation of the serotonin and dopamine neurotransmitter pathways in the developing brain, currently associated with online activity abuse and/or internet addiction, and akin to that found in severe substance abuse syndromes. A general functional working model is proposed under the light of evidence brought to the forefront in this review.

## 1. Introduction

With rapidly spreading digitalization worldwide, more and more people and, in particular, increasingly younger children are spending an increasing number of hours per day online reading on the screens of computers, tablets, and smart phones. This trend has been signaled to now include even the youngest, from age 0 to 2 [1]. Results from several studies suggest that this growing habit is likely to engender multiple health risks such as early myopia and blindness [2,3,4,5,6,7,8,9,10], obesity [11,12], sleep disorders, anxiety, and depression [13,14,15,16,17,18], leading to impaired performance at school and behavioral problems [19,20]. The potential impact of these health risks on our children’s future lives and the well-being of future societies as a whole could be dramatic and public awareness of this problem needs to be fostered in communities as well as on a worldwide scale. This review of environmental studies, metabolic research, and state of the art research in neurobiology points towards the reduced amount of natural day and sunlight exposure of the developing child, as a consequence of increasingly long hours spent indoors online, as the single unifying source, or common denominator, of all the health risks already statistically identified in the literature. In Nordic countries, for example, daylight hours are naturally limited during the winter and this limitation *per se* was found to have a measurable limiting effect on children’s activity levels. In Europe and Australia, evening daylight was found to play a causal role in increasing children’s physical activity [21]. The reported average increases in activity are small but significant and applied to all children of a population and across populations and it was pointed out that additional daylight saving measures may therefore yield public health benefits [21]. In terms of geographic differences [22], children from Melbourne, Australia, and Northern Europe were found to better maintain their activity levels under seasonal changes compared to those in the US and Western Europe [22]. Living in urban or rural environments may also influence children’s levels of physical activity and sedentary behavior, as shown by results from a cross-sectional study on children aged 10–11 in Scotland [23]. Rural children spent an average of 14 min less sitting and 13 min more being moderately active per day compared with urban children from the same country [23].

This review discusses facts and figures to highlight that lack of exposure to natural daylight environments during childhood and adolescence is a direct consequence of worldwide internet penetration and massive digitalization. This represents a tangible risk to children’s and adolescents’ physiological, psychological, and cognitive well-being. Scientific evidence showing that insufficient daylight exposure triggers a chain of interdependent, negative, and potentially long-term effects on the regulation of a child’s physiology will be brought forward. This includes the deregulation of brain mechanisms ensuring far vision, or long-distance visual capacity, vitamin D levels, and neural circuitry, in particular the serotonin and dopamine transmitter pathways, in the still developing brain. Already identified risks of massive digitalization to the health of children and adolescents are brought forward in the first part of this paper. Complex causal links between different metabolic factors involved in a cascade of cause-effect chains, from digital environment to brain function and behavior, are then brought to the forefront. The analysis leads to the conclusion that the health of future generations may be severely compromised if nothing is done to raise public awareness about the necessity for regulatory measures at individual and institutional levels that will effectively prompt children to change and self-monitor their interactions with digital environments wherever possible. This should help minimize some of the risks identified here. The review also clarifies that further research into the direct effects of total digital screen time per day on childhood myopia, sleep disorders, depression, and internet or video game addiction in children and adolescents is urgently needed to facilitate evidence-based policy making worldwide. The standpoint of the author of this review is that society should not wait for risks to be “proven beyond any doubt” to trigger action as at such a late stage, the toll already taken will be higher than anticipated.

## 2. Materials and Methods

The present work is an exploratory focused review (Figure 1) in terms of the Stanford University standards and recommendations for scientific literature reviews [24]. The state of the art research on the topic discussed here is neither mature enough, nor is there a currently existing theoretical framework for performing a systematic quantitative or qualitative meta-review (Figure 1) on the topic with *a priori* defined (i.e., theory-driven) criteria for study inclusion/exclusion. The selection of some of the studies discussed in this review might refer to the author’s personal opinion and/or interpretation of published data and documents reviewed. The major bibliographic research for this paper was performed using a topical search strategy with multiple keyword combinations, exploring the international medical science database MEDLINE using Pubmed, which is hosted by the National Institute of Health (NIH) of the United States of America. The keywords used for this research are given above under “keywords”. This search strategy permitted the selection of topically relevant references to original research articles and reviews retrieved via Pubmed from the MEDLINE database. Further search strategies using the additional keywords “internet use”, “internet penetration”, “worldwide”, “children”, “health”, and “digital environments” and combinations thereof were employed, exploring “google” for topically relevant, openly accessible statistics and/or institutional reports. This search strategy yielded documents available online [25,26], including public reports by the World Health Organization, the European Commission, and the Organisation for Economic Co-operation and Development OECD.

## 3. Results

The goal of this bibliographic analysis was to provide an educated and unifying account of the multiple risks represented by excessive exposure to digital environments to the physiological, psychological, and cognitive health of children and/or adolescents. The review here is written in a narrative style and provided under the working hypothesis that these risks will increase further as children’s overexposure to digital environments increases further. This appears to be the current trend, as will be shown further in this review. While a reasonable and well-balanced use of computers and digital media by children may contribute positively to the development of academic, cognitive, and social skills, such benefits are severely overshadowed by the tangible risks of an excessive exposure to digital environments as revealed by the large majority of facts and figures accounted for in this review. The concerns expressed herein, and the model proposed at the end, acquire their full significance under the light of publicly available population statistics [1,27,28,29,30] showing an alarming trend towards increasingly younger children spending increasingly long hours indoors, invisibly “tethered” to digital devices [27,28,29], and a steep increase in adolescent suicide rates in the US and possibly other countries for which no such statistics are publicly available.

### 3.1. Health Risks Related to Children’s over Exposure to Digital Environments

In the middle ages, access to written sources like scrolls or bound manuscripts was the privilege of clerics and aristocracy. Books were written and copied by hand, then essentially by monks in monasteries, until Gutenberg revolutionized printing and made books available to all. People worldwide started reading and over centuries, shortsightedness caused by reading for long hours under poor light became more and more frequent [25,26]. Subsequently, people learnt that it was important to read under good lighting conditions and to take breaks. Now, there is a new boost in myopia worldwide as a consequence of digital technology. Computers, the internet, smart phones, tablets, and e-books have re-shaped our reading and learning habits entirely and since 2014, there are officially more mobile devices than people in the world [26]. We can access hundreds of resources on almost any subject with just a click of a button and by using search engines, which is much more practical and efficient. Technology has also changed the way we use libraries. Instead of going outdoors to make our way to the library and search the shelves, we reserve the book we want online before fetching it at the library. While this saves us a lot of time, it has also conditioned us, and our children, to spend many more hours indoors reading small text on smaller and smaller screens online. Schools and universities around the world now offer tuition online and for many pupils and students, using a book, pen, and paper has become a thing of the past. Tablets that hold the contents of hundreds of books, video classes, online learning, online homework sessions, and online exams enable students to stay connected with their schools 24 h a day. According to data from the Global Web Index Survey, published online in 2019 [27], the average time spent by individuals between 16 and 64 years of age may be estimated to seven hours per day worldwide (Figure 2). In the USA, in 2017, this number was estimated to amount to two hours and 20 min per day for children between zero and two years of age [28] and to four hours and 30 min for kids between eight and 12 years of age [28,29].

A pilot study was conducted in the framework of the European Union EU project ECIT (Empowering Citizens’ rights in emerging Information Technology). ECIT was aimed at identifying new threats to children as a result of digital technologies beyond social networks to develop recommendations and to empower children’s rights by preventing and mitigating risk issues through education, school/community co-vigilance, and a reorientation of digital and personal interactions through raised awareness and parental guidance. Research has mainly targeted 9–16 year olds, but some have shown that children are going online at an increasingly young age [1]. Very young children cannot be made aware of the risks they incur when gaming online, and despite the substantial increase in online activity in very young children, research is lagging behind. This is alarming and points toward an urgent need for more research fast, as this growing habit may promote the early onset and speedy progress of axial myopia, which ultimately leads to blindness if it is not treated as early as possible.

#### 3.1.1. Myopia and Early Blindness

The evidence for a dramatic rise of myopia, especially in children [2,3,4,5,6,7,8,9,10], sends out a severe warning signal to governments, parents, and clinicians worldwide. Especially in East and Southeast Asia, childhood myopia has risen dramatically in the last 60 years [2,3,4,5,6,7,8,9,10], as extensively documented in reports and studies which describe and analyze the history, epidemiology, and the presumed causes of the worldwide “myopia boom” [3]. This dramatic evolution is linked to the general society trend where adults spend a large part of their time online and where children start out way too early in life looking at the screens of computers, tablets, and smart phones for longer and longer hours every day. In a systematic review article on the association between digital screen time and myopia, the authors conclude that the hours spent on digital screen time in children and adolescents and odds of myopia “are mixed” [30] and that the impact of screen time on myopia has to be further evaluated using objective screen time measures. Myopia prevalence appears to have increased primarily with increasing education in urban Asia a few decades ago and not just recently alongside screen time [30]. There is currently no clear knowledge regarding how many hours per day exactly screen time in children has increased in Asia over the years, but publicly available statistics showing that internet use has grown by a factor of 30 on average in Asia over the last 20 years of the critical reference period [31] suggests that children’s screen time is likely to have increased just as dramatically in the last 20 years. There is strong evidence from other research, presented in a comprehensive overview [32], that lack of exposure to outdoor light is the major cause of the rapid rise in childhood myopia, in Asia and beyond. The lack of exposure to outdoor light as a direct consequence of increased digital screen time in the genesis of early childhood myopia urgently needs to be investigated further in rigorously systematic studies, as suggested in [30]. The specific form of early myopia identified in children is due to an excessive growth of the eye in the longitudinal direction [2] and is referred to as axial myopia [2,32]. If left untreated, the disease progresses, leading to severely impaired vision and ultimately, blindness. In East and Southeast Asia, about 95% of the population needs glasses or contact lenses to restore functional clear vision beyond an arm’s length. This statistic [2] exceeds by far the estimates from an earlier report, published four years ago [3], of an expected increase in myopia prevalence in different countries including Asia between the years 2000 and 2050 (Figure 3). It can be expected that current statistics from other countries also already exceed the predictions made for 2050 in this earlier report.

A likely explanation for this worrisome trend towards rapidly increasing myopia worldwide is that children may now become myopic early in childhood because their eyes grow too fast as a result of excessive time spent reading close-up on increasingly smaller screens of digital devices (computers, tablets, smart phones). This means that children do not get enough outdoor activity and suffer from a cumulated lack of sufficient amounts of daylight. Myopia is estimated to currently affect 108 million people worldwide and is identified as the second most common cause of global blindness [25]. The worldwide economic burden of uncorrected distance vision impairment, of which myopia is the main cause, is currently estimated to 202 Billion Dollars per annum. With the rising prevalence of myopia in younger children, this economic burden will also rise. In addition, myopia is associated with other complications such as myopic retinal detachment, cataract, glaucoma, and macular degeneration. Once myopia has set in, treatment must be initiated as early as possible to stop the progression towards total blindness.

#### 3.1.2. Obesity

Excessive online activity has recently been associated with a significantly higher Body Mass Index (BMI) in pre-adolescent children [11,12], pointing towards a link between digitalization and childhood obesity. Current statistics on childhood obesity collected by the World Health Organization Commission on ending childhood obesity [33] reveal that the number of overweight or obese infants aged 0 to 5 years has increased from 32 million in 1990 to 41 million in 2016. This number is currently projected by the WHO to reach 60 million worldwide by 2035 and ensuring that our kids follow the right diet will not be enough to prevent this from happening. Obesity is deemed one of the most challenging public health problems faced by developed and developing countries worldwide. Screen media exposure is deemed a well documented cause of obesity in children and obesity a well documented consequence of screen media exposure [34,35,36,37,38]. Exposure to “screen media” in this context refers to exposure to content on any screen including that of tablets, computers, smart phones, or TV sets.

Evidence for a direct link between the severity of childhood obesity and the number of screen time hours comes from a cross-sectional survey study within the Childhood Obesity Multi Program Analysis and Study System (COMPASS) on consecutive patients seeking treatment at five tertiary care weight management programs located within geographically diverse children’s hospitals of the United States [36]. Results from this survey indicate that the severity of a child’s obesity increases with the number of hours of daily screen time (Figure 4). To explain the link between screen time and obesity in children, previous theories used the assumption that excessive screen time reduces time spent being physically active and as a consequence, the child will gain weight. However, epidemiologic studies on the one hand point towards much more complex causal links [39,40] and experimental studies on the effects of reduced screen time on measurable increases in physical activity did not yield conclusive results [36,38]. This suggests that the lack of physical activity is not a self-sufficient direct link between long screen times and obesity. There is stronger evidence for increased energy intake as a prominent causal link between screen times and childhood obesity. Epidemiologic studies have shown that children who had more screen time statistically consume fewer fruits and vegetables and more energy snacks, soft drinks, or fast food and therefore receive a higher percentage of their energy from fats and have a higher total energy intake [37].

Eating while viewing increases children’s daily energy intake [35], as demonstrated by studies where children were reported to consume a large proportion of their daily calories and meals during screen time. However, the reasons for this change in the eating habits of the youngest who are exposed to longer screen times are only beginning to be unraveled. They relate to complex metabolic changes involving a variety of factors that will be made clear herein.

#### 3.1.3. Sleep Disorders, Anxiety, Depression

Long hours of exposure to digital technology or online activity have been associated with loss of sleep and/or symptoms of depression in young students [13]. When children or students have to get up early for school or college, delayed bedtimes due to online reading until late can take a serious toll. Several studies have linked delayed bedtimes to poor performance in school, impaired learning, and psychological problems [18,19,20]. Electronic media, especially when used before bedtime, have a negative impact on the sleep of children and adolescents [41,42,43,44,45,46,47,48,49,50,51,52,53,54,55] and while shorter total sleep times have been consistently related to media use, coherent brain-behavior models of the underlying mechanisms are still lacking. However, it has become quite clear now that over exposure to digital environments and the metabolic changes that this produces in children have a measurably negative impact on their cognitive development [56]. Significant relationships between digital device exposure and sleep variables tested in different studies include shorter times spent in bed and shorter total sleep times [43,44,46], delayed bedtimes or longer sleep onset latency, more frequent night waking, delayed wake-up times [45,48,49,50,51,52], daytime sleepiness or tiredness, bed time resistance, sleep anxiety [14,15,16,17,18,19,20,21], sleep-disordered breathing pathologies, sleep–wake transition disorders [19,20], and excessive daytime somnolence [44,45,46,47,48,49,50,51,52,53,54].

Populations tested range from children aged 5 to adolescents aged 19. Virtually no data are available on the effects of the duration of exposure to digital environments, which have been estimated to 2.5 h per day on average for the youngest aged between 0 and 2 years of age in the USA. Partial evidence for negative physiological and psychological effects of excessive exposure to digital environments in children younger than 5 years of age is available [56,57,58], but more research is urgently needed. Most importantly, maladapted and excessive use of the internet results in a new syndrome, now officially referred to as Internet Addiction (IA) or Internet Addiction Disorder (IAD). This syndrome [59] most widely, and severely, affects young people and is associated with various other negative consequences on health and behavior, including sleep disorders, depression, and cognitive dysfunction beyond, as we will see in the next sub-chapter. Large survey studies, behavioural experiments, and molecular and/or functional imaging approaches have been employed over recent years to extend the analysis of the neurobiological changes as a consequence of excessive online activities and to pin down the major neurochemical correlates of internet addiction.

#### 3.1.4. Digital Addiction

Internet Addiction Disorder (IAD) is a disabling condition that calls for full consideration as it has a severe impact on young people’s brain function. Internet addiction disorder (IAD), sometimes also called pathologic/problematic internet use (PIU), is widely defined in terms of an impulse control disorder characterized by uncontrolled Internet use [59,60,61,62,63,64,65,66,67,68,69,70,71,72,73,74,75,76]. The disorder is associated with significant functional impairment and/or clinically measurable distress, anxiety, depression, and other psychopathological symptoms [59,60,61,62,63,64,65,66,67,68,69]. IAD is not (yet) classified as a mental disorder in the Diagnostic and Statistical Manual of Mental Disorders—Fifth edition (DSM-5), but a subtype of IAD, internet gaming disorder (IGD), which specifically refers to videogame addiction, has been included in Section 3 of the DSM 5 [69]. It is currently envisaged to include IAD and IGD also in the International Classification of Diseases for mortality and morbidity statistics ICD-11 [70]. A meta-analysis on IAD performed six years ago [63] and involving more than 89,000 participants from 31 nations reported a global prevalence estimate for IAD of 6% worldwide. The highest estimates for IAD prevalence were scored for the Middle East in terms of about 12% of the reference population, the lowest for Northern and Western Europe, with about 2.5% of the reference population. These estimates were made six years ago. A study conducted on Indian college students [64] identified male gender, continuous online availability, and predominant use of the internet for new friendships/relationships as major risk factors. Higher computer skills and easy Internet access in teenagers and young adults represent an augmented risk for IAD [63,64,65]. Internet addiction (IA) has emerged as a universal issue, but its international estimates vary considerably.

Two factors have been considered to explain cross-national variations [60,61,62,63,64,65]. One is internet access, which varies between continents and nations and predicts that IA prevalence should be positively related to the internet penetration rate per capita. The other factor, referred to as real life quality, predicts that IA prevalence should be inversely related to the global national index of life satisfaction and/or other specific national indices of environmental and lifestyle quality. Personal Technology Usage (PTU) has hit young people in the USA “fast and hard”, with 92% internet penetration, which is currently the highest in the world, as pointed out in an article on the effects of PTU on children and youth [76]. This recent online article published by the US Naval Institute Proceedings describes the internet in terms of a virtual hypodermic mechanism that delivers a digital drug content in a highly effective manner, particularly via the smartphone. This drug seems to hamper children’s ability to manage and balance time, energy, and attention and thereby leads to lifestyle changes and behavioral deficits. The mediating effects of insomnia and associations between problematic Internet use including Internet Addiction (IA), Online Social Networking Addiction (OSNA), and depression among adolescents [58,59,60,61,62,63,64,65,66,67,68] have been highlighted in a population of more than one thousand secondary school students from Guangzhou in China [68], which has about 70% internet penetration, i.e., about 30% less than the USA or Europe. Levels of depression, insomnia, IA, and OSNA were assessed using the Center for Epidemiological Studies Depression Scale (CESDS) [77], the Pittsburgh Sleep Quality Index (PQI) [78], Young’s Diagnostic Questionnaire for Internet Addiction (YDQIA) [79], and the Online Social Networking Addiction Scale (OSNAS) [80]. The results from this cross-sectional study reveal that a high prevalence of IA and OSNA is associated with an increased risk of developing depression among adolescents, both directly and/or as a consequence of insomnia associated with the addictive behaviour [67,68]. Insomnia, therefore, is a factor that may be a trigger for or a chronically developing consequence of IA and OSNA. Likely, depression predicts IA and OSNA, and vice versa, among subjects who were free from either problem at baseline [68]. Conclusions from this study suggest that it may be effective to consider problematic Internet use, insomnia, and depression jointly as all three seem to be clearly interdependent in light of these and other findings [68,69]. The high incidence of depression and increasing suicide rates in teenagers has become a worldwide concern that calls for urgent scrutiny. Depressed individuals might go into online social networking as a secure and non-threatening means of communication with the outside world and as a means for alleviating anxiety related to personal problems. Thus, excessive internet use appears to be a maladapted coping strategy [76] that accelerates the development of digital addiction on the one hand and augments the withdrawal from interpersonal offline activities that could lead to effective coping with real world problems. As a consequence, the young person, instead of coping or learning to cope with problems through real-world interaction with others, spirals further and further down the slippery slope of depression, insomnia, and ultimately, isolation and loneliness. A study published in 2017 by the American Journal of Epidemiology [81] has shown that teenagers are particularly vulnerable in this respect and that the trend towards online technology-induced teenage depression and associated symptoms may well reach epidemic proportions if nothing gets done to stop it. The national statistics made available in a US Naval Institute Publication [76] show estimates for suicide rates in individuals for two age ranges across the years 2000–2019 in terms of % of change per annum. The curves reveal an alarming trend towards an increase, which has been particularly steep for ages 15–34 between the years 2008 and 2016 (Figure 5). The publication explicitly links this trend to digital addiction.

### 3.2. The Early Deregulation of Neurotransmitter Pathways in the Child’s Developing Brain: Towards a Unifying Model Account

What science needs most now is a working model that provides a unifying account of brain-behaviour function [71] underlying the health issues identified here. These appear quite clearly intertwined and may well be the consequence of one and the same environmental variable: excessive exposure to electronic device technology. Molecular and functional imaging findings on neurobiological mechanisms of internet addiction, focusing on magnetic resonance imaging (MRI) and nuclear imaging modalities including positron emission tomography and single photon emission computed tomography, have become available [72,73,74]. MRI studies reveal structural changes in the frontal cortex associated with functional abnormalities in Internet addicted subjects [73]. Nuclear imaging findings indicate that IA is associated with dysfunction of the brain dopaminergic systems [72,73], indicating that de-regulation of the prefrontal cortex may underlie reward specific uncontrolled behavior in the internet overuse of addicted subjects.

Results from a set of independent studies [82,83,84,85,86], mostly conducted in East Asia on young male subjects with internet gaming disorder, were analyzed in a comprehensive metareview [87]. This analysis led to a conclusion on functional alterations, similar to those observed in substance abuse, in brain regions involved in cognitive control functions [82,83,84,85,86,87], and reward/punishment sensitivity balance [84]. These findings connect with other functional evidence from neurobiology, as will become clearer below. The narrative here will highlight the complex cause-effect chain that links increased exposure to digital environments to decreased exposure to healthy natural daylight and increased exposure to artificial light sources at the wrong time of day, deficient vitamin D and melatonin levels in the body, perturbed circadian rhythms as a consequence, and ultimately, the progressive deregulation of the serotonin transmitter pathways in the human brain. This generates a newly emerging syndrome of cognitive and emotional dysfunction akin to that found in severe substance abusers. It will be highlighted how these deregulatory mechanisms are triggered and progressively develop into a syndrome in children, now including even those aged 0 to 2 [58] as a consequence of over exposure to digital environments.

#### 3.2.1. Exposure to the Wrong Kind of Light at the Wrong Time

Myopia is estimated to globally affect 108 million people worldwide and is identified as the second most common cause of global blindness. Further, the worldwide economic burden of uncorrected distance vision impairment, of which myopia is the main cause, is currently estimated at 202 billion dollars per annum [88,89,90]. With the rising prevalence of myopia in children, this economic burden will also rise [89,90,91]. In addition, myopia is associated with other complications such as myopic macular degeneration, retinal detachment, cataract, and glaucoma. Once myopia has set in, prismatic or bifocal lenses, and specially designed multifocal soft contact lenses, and outdoor eye exercises have shown positive results in slowing myopia progression [92,93,94,95,96], which ultimately leads to blindness. Lifestyles that place emphasis on sports and outdoor living and where kids grow up accordingly, as in countries like Canada, Australia, or New Zealand, may explain why these countries have the lowest occurrence of myopia in the world. When kids are spending time outdoors, they are actively *using* and training their long distance vision by focusing on objects further away in their visual field. This is even more critically important in very young children (aged zero to two) with still developing visual systems and brains. The development of visual capacity in very young children takes place over many months [97].

Although current research into the brain development that may limit visual function at an early age suggests a relatively mature neural organization in human infants, despite such anatomical maturity, there is a high degree of visual plasticity with critical sensitive periods [98] of visual functional maturation, highlighted by a differential time-course for the development of form and motion sensitive visual capabilities in normally developing children [99]. It has been suggested that during a critical period, children are particularly vulnerable to any abnormal visual experience [97]. The growing trend in even the youngest to spend their time indoors with their eyes glued to the screen of a smart phone or a computer may, indeed, qualify as an “abnormal visual experience” and prevent them from exercising their far vision capacities under well-balanced natural viewing conditions, as did those from times before the digital age. Like our muscles, our brain and visual capacities tend to weaken when we do not use them properly, especially during functional development in childhood. Results from clinical studies examining the association between hours spent outdoors and prevalent myopia, incident myopia, and myopic progression [93,94,95,96] produced pooled odds ratios and 95% confidence intervals for each additional hour spent outdoors per week from a meta-analysis. The pooled ratios indicated that odds of myopia reduced 2% per additional hour of time spent outdoors per week [93,94]. Prospective cohort studies provided estimates of the risk of incident myopia and myopic progression as a function of time spent outdoors, indicating that increasing time spent outdoors significantly reduced both the risk of incident myopia and the speed of myopic progression. Daily exposure to very bright light per se might protect kids from developing near-sightedness [2]. The known mitigating effects of time spent outdoors towards stopping the progress of childhood myopia [95,96] may partly be due to the fact that outdoor play allows practicing far vision. What is known beyond doubt is that a sufficient amount of exposure to daylight is critically important to preserving good health and minimizing the risks of developing depression and other mood disorders [100].

The temporal organization of human physiology in terms of circadian rhythms [100,101,102,103,104] is critical to our health and especially that of our children. Since electric light was invented, however, pervasive exposure to artificial light sources at night has blurred the boundaries between day and night, making it more difficult for humans to synchronize all sorts of biological processes. Many physiological systems are under the control of circadian rhythms, which influence our sleep–wake behavior, hormone secretion, cellular functions, and even gene expression. Circadian disruption by nighttime light perturbs all those processes and is associated with an increase in the incidence of cancer, metabolic dysfunction, and mood disorders [101,102,104]. Electronic tablet computer screens and smartphone screens can emit more than 40 Lux, depending on the size of the screen. More and more children (and their parents) leave electronic devices such as computers switched on in their bedroom while sleeping [101]. As a consequence, the amount of artificial light exposure at night is, indeed, unprecedented in human history.

Exposure to light at night perturbs the circadian system because light is the major entraining cue used by the body to discriminate day and night and when exposure to light is not timed properly or becomes constant, biological and behavioral rhythms are desynchronized, which has severe consequences for a child’s (and an adult’s) health. Excessive daytime sleepiness as a result of a perturbed day-night rhythm has, indeed, been linked quantitatively to obesity, anxiety, and sleep disorders in young children [105,106,107,108,109]. With the widespread use of portable electronic devices and the normalization of screen media devices in the bedroom, insufficient sleep is now affecting 30% of toddlers, preschoolers, school-age children, and the majority of adolescents [110]. In literature reviews of studies investigating the link between youth screen media use and sleep, 90% of included studies found an association between screen media use and delayed bedtime and/or decreased total sleep time [48,49]. Proposed mechanisms include displacement of time that would have been spent sleeping, psychological stimulation due to blue light source exposure in the evening or even at night, and increased physiological alertness [101]. This pervasive phenomenon of pediatric sleep loss has widespread implications due to the associations between insufficient sleep and increased risk of childhood obesity [105,106], disrupted psychological well-being [105,106,107], and impaired cognitive function [56,110,111]. There is a clear need for more research on the effects of screen media on sleep loss and health consequences in children and adolescents on the one hand and a need for more general information to motivate society stakeholders to foster healthy online behavior to ensure healthy sleep habits. Indeed, as reviewed in the previous chapter above, the habit of spending longer and longer hours reading online has recently been associated with childhood obesity, loss of sleep, and/or symptoms of depression in increasingly younger individuals, and sleep disorders and insomnia in young individuals have both been linked to online addiction. Thus, a holistic analysis of the current evidence reviewed here points towards higher risks of early myopia, obesity, sleep disorder, depression, and online addiction in children who spend too much time online exposed to artificial light sources, often at the wrong time of day, and who lack exposure to a sufficient amount of outdoor daylight.

#### 3.2.2. Resulting Functional Consequences of Vitamin D and Melatonin Deficiency

The amount of daylight children get while they develop is a critically important factor to their healthy development. In our current digital society, daylight exposure is probably reduced to insufficient rates worldwide and this is likely to be more true as digitalization progresses further. Exposure to daylight is essential to the regulation of vitamin D and melatonin production in the human body [112,113,114], as both vitamin D and melatonin ensure important and closely related metabolic functions in the regulation of eating habits and sleep [115,116,117,118,119,120,121,122,123]. Vitamin D helps delay age-related changes in the human body, including degenerative changes in the visual system [124]. Knowing that the outer retina has the highest metabolic demand in the body, retinal health is also dependent on sufficient levels of vitamin D and melatonin in the body [124,125,126,127]. Exposure to as much daylight as possible contributes to healthy vision and may play a hitherto unsuspected role in the prevention of early vision loss in children exposed to digital environments by preventing the accelerated ageing of their retina [125,126]. Exposure to daylight increases levels of retinal dopamine in the visual system of myopic kids and slows down the progression of myopia [2]. The right amount of daylight, especially sunlight, helps produce adequate levels of vitamin D in the human body and a well-regulated rhythm of daylight and nightlight exposure helps produce adequate levels of melatonin.

Since vitamin D and melatonin are functionally related, insufficiency in both engenders health risks [128,129,130,131,132,133,134,135,136,137,138] such as obesity [129], poor sleep [120,121], depression [128,130,133], and addictive behaviours [122,129] well beyond formerly identified physiological issues related to poor bone growth and muscle function [131,132,133]. There is now growing evidence that vitamin D ensures the healthy function of the neurotransmitter dopamine in the central nervous system [128,129,130], leading to new insights into the importance of vitamin D to the health of children beyond the mere recognition of its importance for calcium homeostasis and bone growth [132,133]. Vitamin D significantly impacts the immune systems in charge of preventing infections and regulating autoimmunity [138]. The neurohormonal effects of vitamin D and melatonin deficiency on brain development and behavior, linked to cognitive impairment and mental health disorders [128,129,130,134,136], highlight the interdependency between their metabolic regulation and the regulation of the dopamine neurotransmitter pathways in the brain [134,137,139,140,141,142,143]. The health consequences of vitamin D deficiency include the development of symptoms of dementia due to an increase in cerebral soluble and insoluble peptides and a decrease of its anti-inflammatory/antioxidant properties in the brain [142]. The reduction of buffering of increased calcium in the brain also may cause hypoxic brain damage and promote the development of depression, borderline schizophrenia, and other mental illnesses [128,132]. The fact that obesity rates in children have risen dramatically worldwide over the last years may correlate with low levels of circulating vitamin D3 in their body, which significantly correlates with a high adiposity index [135]. The mechanism of exactly how vitamin D contributes to/interacts with melatonin production and healthy sleep are just beginning to be elucidated. It appears that it might have something to do with vitamin D regulation of tryptophan hydroxylase (TRPH) expression—the rate-limiting enzyme in serotonin and consequently, melatonin production [144]. Vitamin D potentiates the expression of neuronal TRPH to stimulate the appropriate production of serotonin in the brain and without sufficient serotonin production, melatonin levels will not rise appropriately to give the body the signal to go to sleep at night [144,145,146]. Vitamin D regulation in the body is closely linked to melatonin regulation and both are critically influenced by the right amount of light a child is exposed to at the right time of day.

Like vitamin D, melatonin is an antioxidant that scavenges free radicals in organisms and has anti-inflammatory [147], antitumor [148], and antiangiogenic effects [148,149]. Melatonin is a hormone that is mainly produced by the pineal gland when lights are out [137]. A subgroup of photosensitive retinal ganglion cells is responsible for mediating the light-dark cycles that regulate melatonin secretion in the body, and the melatonin release function obeys a circadian rhythm and correlates with sleep patterns [150,151,152,153,154]. Patients with circadian rhythm sleep disorders, including blind patients with no light-induced suppression of melatonin, benefit from melatonin treatment [145,146,150,151]. Melatonin is synthesized in the retina and other parts of the body. Some studies have highlighted the antioxidative, antiapoptotic, and autophagic effects of melatonin on oxidative damage to retinal cells and photoreceptors [147,148,149,150,151,152]. Thus, like vitamin D, melatonin is important to the healthy development of a healthy retina and visual system. Melatonin has been found to be effective in the treatment of insomnia and depression [153,154]. The hormone is produced by the human body at night and like vitamin D, it influences neurotransmitter release functions in the brain [144]. Significant effects of dopamine release under the influence of melatonin have been demonstrated in specific areas of the mammalian brain, and our further understanding of diurnal variation in dopamine is critical for understanding and treating the multitude of psychiatric disorders, including digital addiction, that originate from perturbations of the dopamine system [155]. These elements all taken into consideration suggest that excessive indoor time spent online by children is likely to have a negative effect on a variety of dopamine-dependent behaviors. A pilot study specifically explored the association between peripheral blood dopamine levels and internet addiction disorder (IAD) in adolescents [156], knowing that chemical drug abuse (i.e., cocaine) and pathological gambling have neurobiological effects resulting in increased peripheral dopamine levels. The results of this study showed that peripheral blood dopamine levels in adolescents with internet addiction were twice as high compared with those of healthy controls [156]. This provides further evidence that dopamine pathway regulation is dysfunctional in digital addicts.

#### 3.2.3. Early Deregulation of Neurotransmitter Pathways in the Developing Brain

As discussed above, exposure to daylight directly increases levels of retinal dopamine in the visual system of myopic children and slows down the progression of myopia. Also, as the review has shown above, sufficient amounts of exposure to natural daylight outdoors regulates vitamin D levels, while both the right amount of daylight at the right time and “lights out” at the right time (before going to bed) regulate melatonin levels, knowing that the latter is produced when lights are out and the body stops producing it under daylight exposure. Sufficient levels of both vitamin D and melatonin are necessary for a healthy retina and visual cell function, healthy eating habits, and healthy sleep patterns, as also reviewed in detail above. Both vitamin D and melatonin deficiency caused by excessive exposure to digital environments, especially during childhood, severely interfere with the healthy regulation of the neurotransmitter serotonin in the body [156,157], while the reward circuitries involved in digital addiction change the regulation of the dopamine pathways in the brain [155]. This points towards a tightly interwoven cause-effect chain, linking too much time spent on digital devices indoors on the one hand and digital addiction on the other, to a general deregulation of neurotransmitters involved in the cognitive control of a child’s whole metabolism, from eating habits and sleep patterns to general intellectual capacity.

Serotonin is a neurotransmitter that contributes to the healthy regulation of a large number of physiological processes and its production depends on healthy levels of vitamin D and melatonin. Both melatonin and vitamin D synthesis are affected by light, and vitamin D is directly involved in melatonin secretion [157,158]. Serotonin is both an excitatory and an inhibitory neurotransmitter found in enteric neurons and in the brain [157,158,159]. Serotonin synthesis requires magnesium, zinc, and vitamin B6 and vitamin C as cofactors. In the pineal gland and the retina, the enzyme *N*-acetyltransferase converts serotonin to *N*-acetyl serotonin, which in turn is converted to melatonin and released into the bloodstream and cerebrospinal fluid by the enzyme 5-hydroxyindole-*O*-transferase, a process that requires the active form of vitamin B6 [157]. As a consequence, the serotonin neurotransmitter pathways ensure central brain control of the rhythm of sleep/wake periods and the immune response of the whole organism. This control breaks down in a so-called serotonin-and-melatonin-deficiency syndrome, which is frequently diagnosed in elderly patients [160]. This syndrome may be seen as a correlate of premature ageing and when present in the young, rings a serious alarm bell, indicating that the mind-body system is under severe stress. Serotonin has an important role in decision making behaviour [161,162]. Serotonergic antidepressants and anxiolytic and antipsychotic drugs are extensively used in the treatment of neuropsychiatric disorders characterized by impaired decision making [161]. High serotonin levels are generally associated with improved reversal learning, improved attentional set shifting, decreased delay discounting, and increased response inhibition [161].

The neurotransmitter dopamine is produced in the substantia nigra, ventral tegmental area, and hypothalamus of the human brain. Dysfunction of the dopamine system has been related to a variety of nervous system diseases [163,164]. Dopamine levels in the brain and the periphery (blood) increase in response to any type of reward and to a number of functionally identified chemical substances and/or non-chemical drugs [164], which include sex, gambling, and most recently, the “digital drug” [157]. Dopamine transmitter pathway deregulation is a consequence of oxidative stress in the body [163]. Interactions between the serotonergic and dopaminergic transmitter pathway systems, both at the anatomical and the functional level, have been identified [164]. In mammals, the central serotonergic system modulates the activity of dopaminergic neurons in the circuits connecting the substantia nigra to the striatum and the ventral tegmental area. Substances like reserpine and amphetamine induce symptoms that closely resemble those associated with depression and/or schizophrenia, and the pharmacological treatment of both directly targets the serotonergic and dopaminergic neurotransmitter systems [164]. Both dopamine and serotonin play an important role in drug and alcohol dependence by mediating the mechanism of dopamine reward and withdrawal symptoms. Patients with Internet Gaming Disorder (IGD) show a significant decrease in the level of availability of dopamine in the striatum and a reduced availability of serotonin [165]. This appears consistent with functional magnetic resonance imaging studies showing that IGD adolescents and adults have reduced gray matter volume in regions associated with attention motor coordination executive function and lower white matter measures in brain regions that control for both serotonin and dopamine dependent decision-making, behavioral inhibition, and emotional regulation, leading to increased risk-taking and diminished impulse control ability, which is common in all forms of addiction [166]. All these elements taken into account, a unifying working model account of the whole picture of interdependent early childhood metabolic disorders and health problems resulting from overexposure to digital environments may be proposed (Figure 6).

## 4. Conclusions

This exploratory focused review of critical elements from the current literature shows quite unequivocally that the projection of increasingly excessive time spent online indoors [167] by increasingly younger children is likely to put their physical and psychological development and general health at risk, in both the short and long term. Early childhood myopia, disturbed circadian rhythms, sleep loss, depression, and ultimately, addiction and the deregulation of central control functions in the brain, initiated by lack of exposure to healthy outdoor light conditions, are the main risks identified here. Myopia is estimated to affect 108 million people worldwide and is identified as the second most common cause of global blindness. The worldwide economic burden of uncorrected distance vision impairment, of which myopia is the main cause, is currently estimated at 202 billion dollars per annum. With the rising prevalence of myopia in children, this economic burden will also rise. In addition, myopia is associated with other complications such as myopic macular degeneration, retinal detachment, cataract, and glaucoma. School-based clinical trials have demonstrated that increasing the amount of time that children spend outdoors to a little more than two hours a day can significantly slow the onset of myopia. It may be necessary to implement mandatory outdoor programs in schools and the regular monitoring of visual acuity of children from age two worldwide. Those children who have already become myopic should be referred for clinical treatment as soon as possible to slow down the progression of their myopia. The recent rise in depression or oxidative stress syndrome [168] in teenagers and even younger children is also alarming. As pointed out earlier, scientific studies have linked depression in students to their online behavior. There may be other factors involved as well, however a devoted public campaign aimed at raising levels of awareness worldwide that too much screen time is not only bad for the eyes, but also for the soul of children, teens, and adults, seems to be a good idea from many points of view.

Statistics on childhood obesity collected by the World Health Organization lead one to conclude that the number of overweight or obese infants aged 0 to 5 years has increased from 32 million in 1990 to 41 million in 2016. This is currently projected to reach 60 million worldwide by 2035, and ensuring that our youngest follow the right diet may not be enough to prevent this from happening. While time spent online by our children is likely to increase further, scientific experts think that the currently recommended doses of food supplementation fall way short of what is needed to obtain the necessary levels for optimal health. A worldwide food supplementation program [169,170] seems urgently needed to prevent the progress of vitamin D and/or melatonin deficiency related health problems in children (and also adults). Plant generative organs (e.g., flowers, fruits), and especially seeds, have been proposed as having the highest melatonin concentrations. While reinforcing a worldwide supplementation program could certainly have a dramatic impact on children’s health worldwide, the first step to take is to raise public awareness of the tight link between time spent online indoors and severely compromised long-term brain health of young children and adolescents. Diet control and food supplements can help here, but are clearly not the sole solution. The spreading disease of digital overexposure or abuse [171,172] may have already taken a toll on even the youngest (toddlers between 0 and 2 years of age) worldwide, but is not yet measurable in terms of short and long term consequences on their bodies and brains. If science and society do not take a clear stance, this growing trend may soon have dramatic and potentially irreparable consequences. As pointed out earlier, reasonable and well-balanced use of computers and digital media by children, contributing positively to the development of academic, cognitive, and social skills [173,174,175,176], has its benefits, but these are already severely overshadowed by the many increasingly tangible risks of excessive exposure to digital environments. The selection of some of the studies discussed in this review may partially reflect the author’s personal opinion. The interpretation of the data and documentation reviewed here is that of the author. The message conveyed is that early warning signals are detectable in the current literature and we must take care to get our children off the “digital hook” before it may be way too late and considerable evidence of damage beyond repair accumulates in the scientific literature. In light of the material reviewed here and with the precautionary principle [177,178] in mind, the time to act is now.

## Figures and Tables

**Figure 1 ijerph-17-03240-f001:**
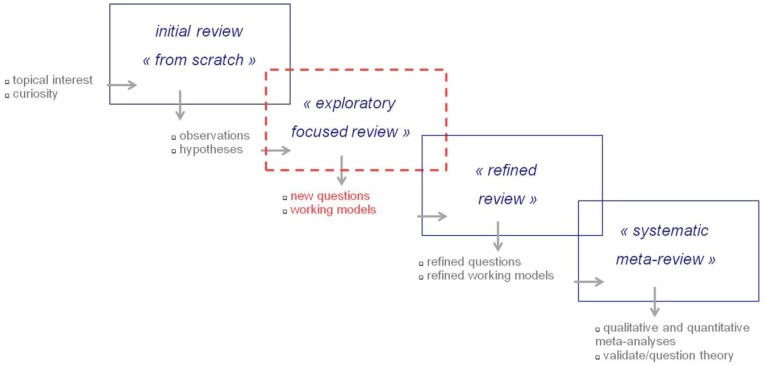
Flowchart of the scientific literature review process, inspired by Liston [24]. A hypothesis-driven, exploratory focused review (cf. the Stanford University standards and recommendations for scientific literature reviews [24]) was performed in this work to produce a temporary working model and inspire further systematic research.

**Figure 2 ijerph-17-03240-f002:**
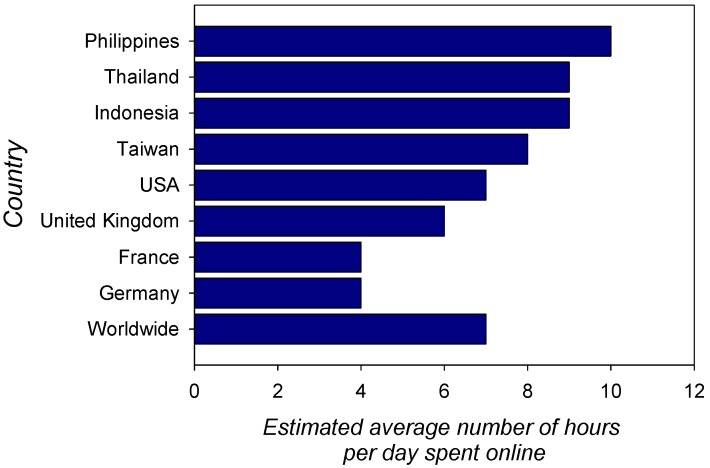
Estimated average time (number of hours per day) spent online by individuals between 16 and 64 years of age [27]. Times vary, as shown here, between countries. The average time spent online is currently estimated to seven hours per day worldwide and is expected to increase further. In the USA in 2017, this number was estimated to amount to two hours and 20 min per day for children between zero and two years of age and four hours and 30 min for kids between eight and 12 years of age [28,29].

**Figure 3 ijerph-17-03240-f003:**
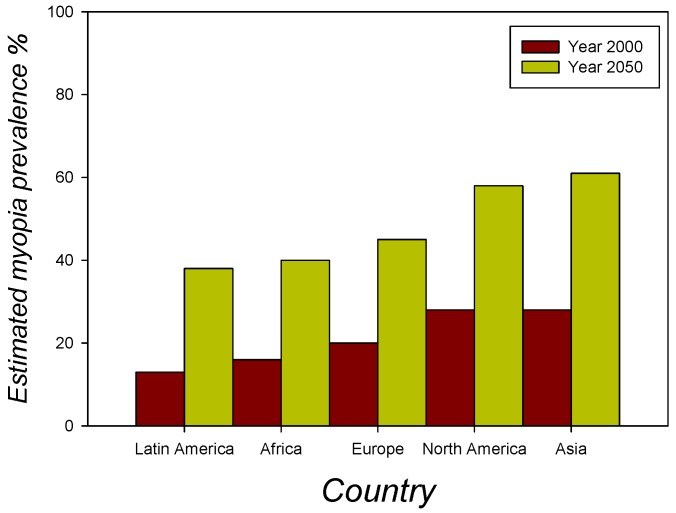
Estimated increase in myopia prevalence in populations from different countries of the world including Asia between the years 2000 and 2050 [3]. The percentages shown here are based on data from an earlier report published five years ago. Current statistics from a more recent review [2] reveal that in 2019, only 5% of the total populations of East and Southeast Asia still had normal (uncorrected) vision. This leads one to suggest that higher percentages than shown above are to be anticipated for the year 2050 if nothing gets done to stop the worldwide trend of myopia.

**Figure 4 ijerph-17-03240-f004:**
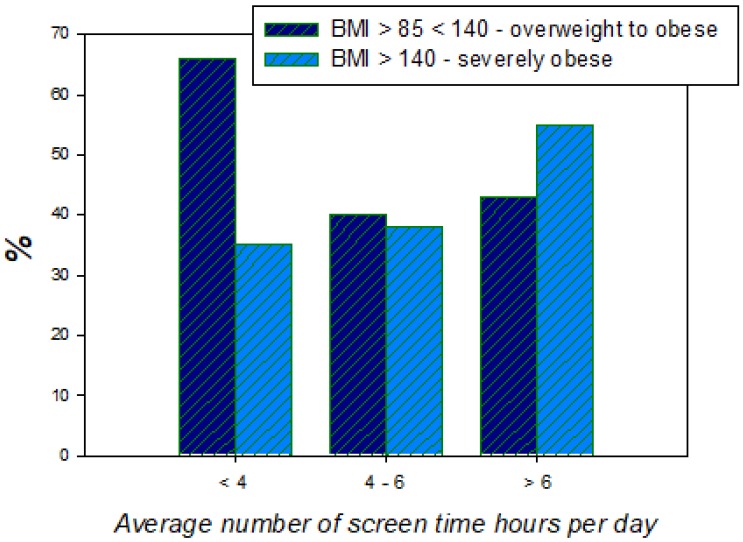
Results of a survey from the USA revealing the link between the severity of obesity and the number of screen time hours per day in children treated for obesity in different hospitals across the country [36].

**Figure 5 ijerph-17-03240-f005:**
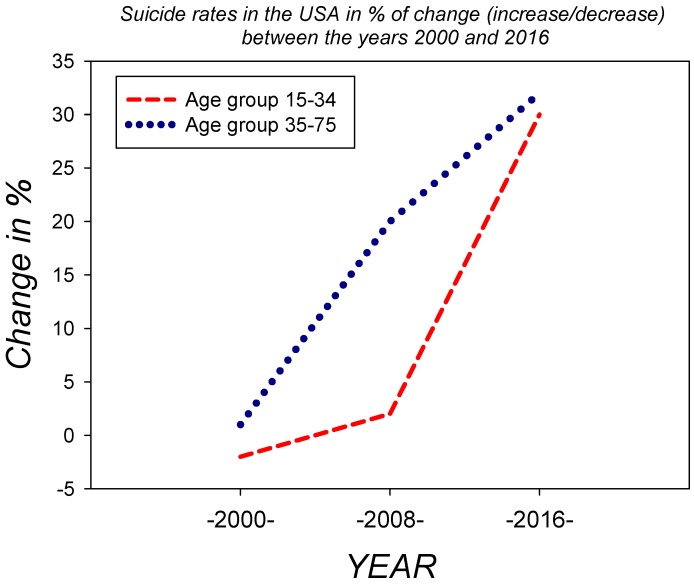
Estimates for suicide rates in the USA, which has an internet penetration index of 92% of the global population, between the years 2000 and 2016. Estimates are expressed herein as the % of change (increase/decrease) per annum, showing an alarmingly steep increase for the age group of 15 to 34 year old individuals. The key data shown here have been replotted on the basis of a report published by the US Naval Institute in 2018 [76], which points towards a link between the increase shown here and digital addiction, especially in younger individuals. The hypothesis is consistent with data from scientific studies, summarized above, showing a tight connection between internet addiction, insomnia, and depression in young males and females.

**Figure 6 ijerph-17-03240-f006:**
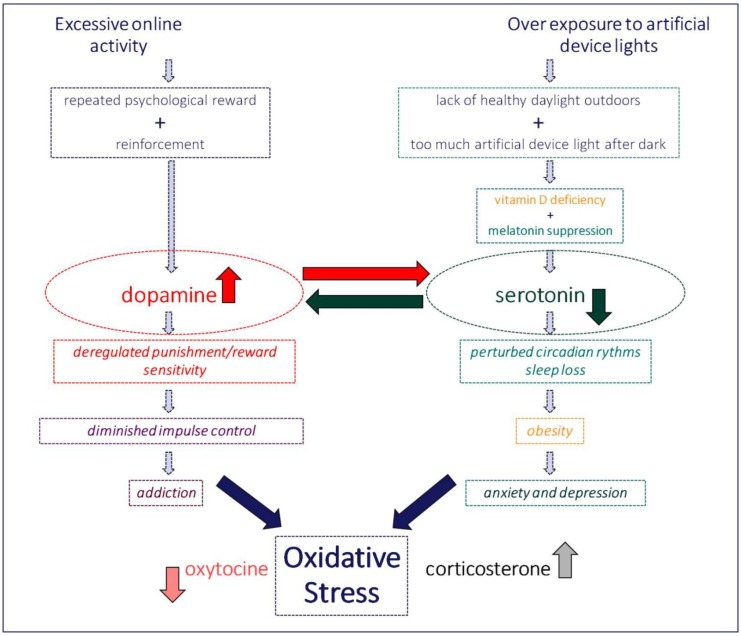
Excessive online activity and, ultimately, digital addiction inevitably go along with an over exposure to artificial device lights. Each of these two produces clearly identified changes in metabolic rates that negatively affect either the dopamine or the serotonin transmitter pathways in the child’s still developing brain. Anticipated long-term consequences of such early brain deregulation are incommensurable. The working model above takes into account evidence from state-of-the-art research studies reviewed in this paper.

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
