# Peer review of "Children’s Health in the Digital Age"

_ijerph, 2020, doi:10.3390/ijerph17093240_

Round 1
Reviewer 1 Report
Thank you for the opportunity to review this paper entitled “Children’s Health in the Digital Age” which was submitted as a review. No details are provided as to what was reviewed and how the source papers were found, and or included. As such, the results and interpretation are unfortunately open to bias, from potentially omitted or overlooked papers and also the chance that the author only presents literature that supports their preferred hypothesis. That being said, it is in effect, an opinion piece backed by data chosen by the author.
I found the paper interesting. As this is a review, the information is not new but the author has joined the dots as it were and presented a model.
Some queries and questions:
Lines 88-92: no references have been provided. The subsequent section on axial myopia lines 94-96 cites several papers reporting “a dramatic rise of myopia and early blindness in children as a consequence of excessive exposure to computer and smartphone screens.” The cited paper on myopia in the United States from 1999-2004, makes no mention of children, computers or phones. Neither does the paper on an adult population in Japan published in 2006. The paper on Taiwanese schoolchildren from 1983-2000 does not mention computers or smartphones as a cause of myopia, neither does the paper on age-specific prevalence of myopia in Asia. These are the only references checked, but the discrepancies noted bring the appropriateness all other references into question.
Introduction, lines 33-38: in Nordic countries, for example, daylight hours are limited during winter. No mention is made of the literature on the effect of this on children.
Figure 1: the reference has not been included.
Figure 4 cannot be interpreted in a black and white printout.
Lines 276-278: are the structural changes cause or effect?
Lines 280-282: can a 2014 paper be considered recent? Its content is, at best, six years old.
Line 287: is there unhealthy natural daylight?
Sections of the paper are repeated; for example, lines 296-300 are the same or present the same information as lines 122-129.
The style is somewhat informal in places, with, for example, repeated reference to kids.
There are a number of spelling errors, typos and minor English errors.
There are several very long paragraphs, several of which are more than a page. This makes reading difficult.
Author Response
Thank you for your most helpful comments and suggestions, which were fully taken into account. The following amendments to the manuscript (all changes are highlighted in red for easy tracking) were made:
- The Reviewer’s comment regarding selection of the papers for this review
I have made the keyword-based search strategies I used to find the documents on which this review is based explicit under ‘Materials and Methods’ (new section) on page 2 after the ‘Introduction’:
“The major bibliographic research for this paper was performed using a topical search strategy with multiple keyword combinations, exploring the international medical science database MEDLINE using Pubmed, which is hosted by the NIH. The keywords used for this research are given here above under ‘keywords’. This search strategy yielded a total of 160 topically relevant references to original research articles and review articles retrieved via Pubmed from the MEDLINE database [2-24,29-31,33-67,70-74,79-169,171-175]. Further search strategies using the additional keywords ‘internet use’, ‘internet penetration’, ‘worldwide’, ‘children’, ‘health’ and ‘digital environments’, and several combinations thereof, were employed exploring ‘google’ for topically relevant, openly accessible statistics and/or institutional reports. This search strategy yielded a total of 15 documents [1,24-28,30,32,68,69,75-78,170] available online, including public reports by the World Health Organization, the European Commission, and the OECD.”
The Reviewer is, of course, right in saying that the paper was written from a certain perspective, or viewpoint, with the goal to provide a unifying model at the end. To clarify this in the article, and from the outset, I have added a short paragraph to the beginning of the ‘Results’ section (new) to 1) explain the rationale of this viewpoint, and 2) to bring forward references to papers that have shown a beneficial effect of moderate computer use on children’s social skills or cognitive development. As the Reviewer will see, the working hypothesis underlying this review paper is based on overexposure to, or excessive use of digital technology by children, not on moderate computer/digital device use:
“The goal of this bibliographic analysis was to provide an educated and unifying account of the multiple risks represented by an excessive exposure to digital environments to the physiological, psychological, and cognitive health of children and/or adolescents. The review here is written in a narrative style and provided under the working hypothesis that these risks will increase further as children’s overexposure to digital environments increases further. This appears to be the current trend, as will be shown further below in this review. While a reasonable and well-balanced use of computers and digital media by children may contribute positively to the development of academic, cognitive, and social skills [172-175], such benefits are severely overshadowed by the tangible risks of an excessive exposure to digital environments as revealed by the large majority of facts and figures accounted for in this review. The concerns expressed herein, and the model proposed at the end, acquire their full significance under the light of publically available population statistics [1, 26-29, 69, 75] showing an alarming trend towards increasingly younger children spending increasingly long hours indoors, invisibly “tethered” to digital devices [26,27,28] and a steep increase in adolescent suicide rates in the US [75] and possibly other countries for which no such statistics are publically available.”
- The Reviewer’s comment regarding the subsection on axial myopia in children and references provided, or lack thereof, on a possible “direct” effect of digital exposure or screen time on childhood myopia
This subsection, on page 4, was carefully and completely revised to ensure that there are no further discrepancies between the references and what is stated in the text. A missing reference to the latest state of the art review on this question (Lanca and Saw, 2020) is now also included in the text and the bibliography:
“Recent evidence for a dramatic rise of myopia, especially in children [2-10], sends out a severe warning signal to governments, parents, and clinicians worldwide. Especially in East and Southeast Asia childhood myopia has risen dramatically in the last 60 years [2-10], as extensively documented in reports and studies which describe and analyze the history, epidemiology, and the presumed causes of the worldwide “myopia boom” [3]. This dramatic evolution is linked to the general society trend where adults spend a large part of their time online, and were children start out way too early in life looking at the screens of computers, tablets and smart phones for longer and longer hours every day. In a systematic review article on the association between digital screen time and myopia, the authors conclude that the hours spent on digital screen time in children and adolescents and odds of myopia are ambivalent [29], and that the impact of screen time on myopia has to be further evaluated using objective screen time measures. Myopia prevalence appears to have increased primarily with increasing education in urban Asia a few decades ago and not just recently alongside with screen time [29]. There is currently no clear knowledge by how many hours per day exactly screen time in children has increased in Asia over the years, but publically available statistics showing that internet use has grown by a factor of 30 on the average in Asia over the last 20 years of the critical reference period [30] suggest that children’s screen times are likely to have increased just as dramatically in the last 20 years. There is strong evidence from other research, presented in a comprehensive overview [31], that lack of exposure to outdoor light is the major cause of the rapid rise in childhood myopia, in Asia and beyond. The lack of exposure to outdoor light as a direct consequence of increased digital screen time in the genesis of early childhood myopia urgently needs to be investigated further in rigorously systematic studies, as suggested in [29].”
- The Reviewer’s suggestion to include references to work on seasonal daylight hours and their effect on children in the introduction.
This excellent suggestions was also fully taken into account, and the following amendment was made accordingly to the introduction on pages 1-2:
“In Nordic countries, for example, daylight hours are naturally limited during the winter, and this limitation per se was found to have a measurable limiting effect on children’s activity levels. In Europe and Australia, evening daylight was found to play a causal role in increasing children's physical activity [21]. The reported average increases in activity are small but significant and applied to all children of a population and across populations and it was pointed out that additional daylight saving measures may therefore yield public health benefits [21]. In terms of geographic differences [22], children from Melbourne, Australia and Northern Europe were found to better maintain their activity levels under seasonal changes compared to those in the US and Western Europe [22]. Living in urban or rural environments may also influence children's levels of physical activity and sedentary behavior, as shown by results from a cross-sectional study on children aged 10-11 in Scotland [23]. Rural children spent an average of 14 minutes less sitting, and 13 minutes more being moderately active per day compared with urban children from the same country [23].”
4.The Reviewer’s remarks relative to spelling mistakes and/or minor issues were all fully taken into accountSpelling mistakes were fixed; “kids” was replaced by “children” in the text; functional and structural changes are, indeed, “consequence” not “cause”; the text sections now have paragraphs marked by indents for easier reading
Reviewer 2 Report
The manuscript is focused on the effect of screen use on children's health and represent an extremely timely and important response to the problem. The manuscript is presented logically with a clear structure. A short description for methodology (selection of the analyzed studies, inclusion and exclusion criteria) would benefit the readers.
Minor editing:
line 125: upper- versus lower case for "Dollar"
line 193: given that this is a manuscript rather than a book, "subsection" may be a better choice instead of "sub-chapter"
Most of the edition is required for the reference section; mostly to ensure consistency in using commas or initials. For example, line 550 includes a comma between authors last name and initial, while line 552 does not, and line 770 starts with author initials.
Author Response
Thank you for your most helpful comments, which were gratefully taken into account as follows:
A short description for methodology, search strategies, selection of the analyzed studies etc. is now given in a ‘Materials and Methods’ section on page 2:
“The major bibliographic research for this paper was performed using a topical search strategy with multiple keyword combinations, exploring the international medical science database MEDLINE using Pubmed, which is hosted by the NIH. The keywords used for this research are given here above under ‘keywords’. This search strategy yielded a total of 160 topically relevant references to original research articles and review articles retrieved via Pubmed from the MEDLINE database [2-24,29-31,33-67,70-74,79-169,171-175]. Further search strategies using the additional keywords ‘internet use’, ‘internet penetration’, ‘worldwide’, ‘children’, ‘health’ and ‘digital environments’, and several combinations thereof, were employed exploring ‘google’ for topically relevant, openly accessible statistics and/or institutional reports. This search strategy yielded a total of 15 documents [1,24-28,30,32,68,69,75-78,170] available online, including public reports by the World Health Organization, the European Commission, and the OECD.”
The paper now also, as a consequence, has a ‘Results’ section; the new structure should benefit the readers.
The references (edits) were fixed, and minor issues corrected to the best of my ability
Round 2
Reviewer 1 Report
Thank you for the opportunity to re-review this paper. Unfortunately, the authors have, not responded to each query.
Previous Query
No details are provided as to what was reviewed and how the source papers were found, and or included. As such, the results and interpretation are unfortunately open to bias, from potentially omitted or overlooked papers and also the chance that the author only presents literature that supports their preferred hypothesis. That being said, it is in effect, an opinion piece backed by data chosen by the author.
Response
I have made the keyword-based search strategies I used to find the documents on which this review is based explicit under ‘Materials and Methods’ (new section) on page 2 after the ‘Introduction’:
“The major bibliographic research for this paper was performed using a topical search strategy with multiple keyword combinations, exploring the international medical science database MEDLINE using Pubmed, which is hosted by the NIH. The keywords used for this research are given here above under ‘keywords’. This search strategy yielded a total of 160 topically relevant references to original research articles and review articles retrieved via Pubmed from the MEDLINE database [2-24,29-31,33-67,70-74,79-169,171-175]. Further search strategies using the additional keywords ‘internet use’, ‘internet penetration’, ‘worldwide’, ‘children’, ‘health’ and ‘digital environments’, and several combinations thereof, were employed exploring ‘google’ for topically relevant, openly accessible statistics and/or institutional reports. This search strategy yielded a total of 15 documents [1,24-28,30,32,68,69,75-78,170] available online, including public reports by the World Health Organization, the European Commission, and the OECD.”
New comment
- The previous comment has only been partially addressed. The search terms should be included in the methods and not merely be referred to as the keywords provided for journal referencing.
- No indication is given as to the Boolean logic used in the search and how the additional search terms were searched. I ran the search in PubMed using the keywords linked with the Boolean operator OR. This returned 3,701,768 papers. When linked with AND there were no papers found. The “multiple keyword combinations” must be given. The method section does not currently provide enough information to replicate the search.
- No inclusion or exclusion criteria are provided
- How many papers were found in the primary search before any exclusions?
- How were the topically relevant papers selected?
- The inclusion of a PRISMA figure describing the search would assist the reader and add value to the paper.
- When was the review performed?
Previous query
Lines 88-92: no references have been provided.
This has not been addressed
Previous query
The subsequent section on axial myopia lines 94-96 cites several papers reporting “a dramatic rise of myopia and early blindness in children as a consequence of excessive exposure to computer and smartphone screens.” The cited paper on myopia in the United States from 1999-2004, makes no mention of children, computers or phones. Neither does the paper on an adult population in Japan published in 2006. The paper on Taiwanese schoolchildren from 1983-2000 does not mention computers or smartphones as a cause of myopia, neither does the paper on age-specific prevalence of myopia in Asia. These are the only references checked, but the discrepancies noted bring the appropriateness all other references into question.
New query
The author has rewritten the paragraph, but the issues raised about the inappropriateness of some references remain unaddressed. Two additional references have been added, the first by Lanca and Saw was only published in March 2020 and so could not have been found in the literature review. The second, to Internet use in Asia is incorrectly dated in the references, it is from 2018 and not 2020.
Previous comments and queries
Figure 4 cannot be interpreted in a black and white printout.
Not addressed
Lines 276-278: are the structural changes cause or effect?
Not addressed
Lines 280-282: can a 2014 paper be considered recent? Its content is, at best, six years old.
Not addressed
Line 287: is there unhealthy natural daylight?
Not addressed
Sections of the paper are repeated; for example, lines 296-300 are the same or present the same information as lines 122-129.
Not addressed
There are several very long paragraphs, several of which are more than a page. This makes reading difficult.
Not addressed
New comment.
There are still several spelling errors, for example, Oxydative instead of Oxidative in figure 5.
Author Response
1 – Reviewer’s previous comment
No details are provided as to what was reviewed and how the source papers were found, and or included. As such, the results and interpretation are unfortunately open to bias, from potentially omitted or overlooked papers and also the chance that the author only presents literature that supports their preferred hypothesis.
1 – Reviewer’s new comment
- The previous comment has only been partially addressed. The search terms should be included in the methods and not merely be referred to as the keywords provided for journal referencing.
- No indication is given as to the Boolean logic used in the search and how the additional search terms were searched. I ran the search in PubMed using the keywords linked with the Boolean operator OR. This returned 3,701,768 papers. When linked with AND there were no papers found. The “multiple keyword combinations” must be given. The method section does not currently provide enough information to replicate the search.
- No inclusion or exclusion criteria are provided
- How many papers were found in the primary search before any exclusions?
- How were the topically relevant papers selected?
- The inclusion of a PRISMA figure describing the search would assist the reader and add value to the paper.
- When was the review performed?
1 – Author’s reply to the new comment (extending my reply to the previous comment)
As already pointed out in my previous reply, but maybe not explicitly and clearly enough, this article is a focused exploratory review based on a selection of high quality peer-reviewed articles and other documents, pointing towards potential risks of digital technology misuse or abuse and/or overexposure to digital screens to the health of children and adolescents. The authors has followed the Stanford University standards and recommendations for scientific literature reviews, which is accessible online at:
https://web.stanford.edu/class/cee320/CEE320A/POD.pdf (last accessed April 2020)
A figure (new Figure 1) was added to the Materials and Methods for further clarification. A focused exploratory review is hypothesis-driven and performed at a stage where no theoretical model(s) exist. An exploratory focused review is generally performed on author-selected literature, and may provide a temporary, early-stage working model to inspire further investigation. The Reviewer will see at a glance at the new Figure 1 why a PRISMA flowchart cannot apply here: the state of the art on the topic discussed is neither mature enough, nor is there a currently existing working model to drive theory-driven systematic meta-review efforts based on an exhaustive screening of bibliographic databases (cf. PRISMA flowchart), with à priori criteria for inclusion or exclusion of data for qualitative and quantitative meta-analyses. To fully address the Reviewer’s comments, I refer to the following additional statements in the Materials and Methods section on page 2. The text highlighted in red the manuscript reflects my previous account of the Reviewers previous comment, the new blue text my reply to the new comment:
“The present work is an exploratory focused review (Figure 1) in terms of the Stanford University standards and recommendations for scientific literature reviews [24]. The state of the art on the topic discussed here is neither mature enough, nor is there a currently existing theoretical framework for performing a systematic quantitative or qualitative meta-review (Figure 1) on the topic with a priori defined (i.e. theory-driven) criteria for study inclusion/exclusion. The selection of some of the studies discussed in this review might refer to the author’ personal opinion and/or interpretation of published data or documentation reviewed here.”
The new text, the new Figure 1, and the author replies here above fully address the Reviewer’s list of remarks with bullet-points. Regarding the last question with a bullet-point: the review was performed between January and April 2020; some additional references were included by the author during the review process.
2- Reviewer's previous and current query
Lines 88-92: no references have been provided. This has not been addressed
2 – Author’s reply to previous and current query
These lines have moved up with the previous and this current revision. The sentence the Reviewer refers to here has been reformulated since and does as such not require a reference.
3 – Reviewer's previous query
The subsequent section on axial myopia lines 94-96 cites several papers reporting “a dramatic rise of myopia and early blindness in children as a consequence of excessive exposure to computer and smartphone screens.” The cited paper on myopia in the United States from 1999-2004, makes no mention of children, computers or phones. Neither does the paper on an adult population in Japan published in 2006. The paper on Taiwanese schoolchildren from 1983-2000 does not mention computers or smartphones as a cause of myopia, neither does the paper on age-specific prevalence of myopia in Asia. These are the only references checked, but the discrepancies noted bring the appropriateness all other references into question.
3 – Reviewer’s new query
The author has rewritten the paragraph, but the issues raised about the inappropriateness of some references remain unaddressed. Two additional references have been added, the first by Lanca and Saw was only published in March 2020 and so could not have been found in the literature review. The second, to Internet use in Asia is incorrectly dated in the references, it is from 2018 and not 2020.
3 – Author’s reply
The recent review by Lanca and Saw (2020) was not available in January, and was added with the previous revisions as this paper directly addresses the question of a link between digital screen time and childhood myopia. The paper concludes that effects are potentially mixed with the effects of other factors, and that more studies with controlled testing of the screen time factor are needed to sort this out. The online reference to Internet use in Asia was last accessed by the author in March 2020, but the article was, indeed, written in 2018. This has been corrected in the references.
4- Reviewer’s previous comment
Figure 4 cannot be interpreted in a black and white printout. Not addressed.
4 - Author’s reply
When I print out this page, I see a clear graph in dashes for one plot, and a clear graph in dos for the other, each symbol (dash or dot) has a clear legend. I do not see any problem of interpreting the black and white printout.
5 – Reviewer’s previous comment
Lines 276-278: are the structural changes cause or effect? Not addressed
5 - Author’s reply
Again, this query was fully addressed in my previous revisions, the typo was corrected and the change highlighted in colour, but the lines had moved up with new text added. The structural and functional changes discussed here are, of course, the effect (consequence), not the cause of the environmental variable under consideration. See lines 345-347 of the revised manuscript, highlighted in blue:
“They appear quite clearly intertwined as consequences of the same environmental variable: excessive exposure to electronic device technology.”
6- Reviewer’s previous comment
Lines 280-282: can a 2014 paper be considered recent? Its content is, at best, six years old. Not addressed
6 – Author’s reply
I agree. This also was addressed in the previous round, but maybe I forgot to list it in detail. The word recent has been deleted in that sentence, and is used more parsimoniously now.
7- Reviewer’s previous comment
Line 287: is there unhealthy natural daylight? Not addressed
7- Authors reply
This also was addressed in the previous round, but maybe I forgot to list it in detail. The word “unhealthy” has been deleted.
8- Reviewer’s previous comment
Sections of the paper are repeated; for example, lines 296-300 are the same or present the same information as lines 122-129. Not addressed
8- Author’s reply
I had revised these sections to the best of my ability to minimize any redundancy in contents, but I did not list this point explicitly in my previous reply.
9- Reviewer’s previous comment
There are several very long paragraphs, several of which are more than a page. This makes reading difficult. Not addressed
9- Author’s reply
I am sorry. I had introduced indents which separate the subsections of each chapter better in the hope this helps.
10 – Reviewer’s new comment
There are still several spelling errors, for example, Oxydative instead of Oxidative in figure 5.
10- Author’s reply
Thanks for pointing this out. Figure 5 (now Figure 6, since a new Figure 1 was added) was amended to correct this spelling error. I have also checked the manuscript text to the best of my ability to ensure that there are no longer any spelling errors.